# Soil Tillage and Crop Growth Effects on Surface and Subsurface Runoff, Loss of Soil, Phosphorus and Nitrogen in a Cold Climate

**Marianne E. Bechmann *** and **Frederik Bøe**

Norwegian Institute of Bioeconomy Research, P.O. Box 115, NO-1431 Ås, Norway; frederik.boe@nibio.no
* Correspondence: marianne.bechmann@nibio.no; Tel.: +47-41219506

**Abstract:** Most studies on the effects of tillage operations documented the effects of tillage on losses through surface runoff. On flat areas, the subsurface runoff is the dominating pathway for water, soil and nutrients. This study presents results from a five-year plot study on a flat area measuring surface and subsurface runoff losses. The treatments compared were (A) autumn ploughing with oats, (B) autumn ploughing with winter wheat and (C) spring ploughing with spring barley (n = 3). The results showed that subsurface runoff was the main source for soil (67%), total phosphorus (76%), dissolved reactive phosphorus (75%) and total nitrogen (89%) losses. Through the subsurface pathway, the lowest soil losses occurred from the spring ploughed plots. Losses of total phosphorus through subsurface runoff were also lower from spring ploughing compared to autumn ploughing. Total nitrogen losses were higher from autumn ploughing compared to other treatments. Losses of total nitrogen were more influenced by autumn ploughing than by a nitrogen surplus in production. Single extreme weather events, like the summer drought in 2018 and high precipitation in October 2014 were crucial to the annual soil and nutrient losses. Considering extreme weather events in agricultural management is a necessary prerequisite for successful mitigation of soil and nutrient losses in the future.

**Keywords:** soil erosion; nutrient loss; ploughing; winter wheat; runoff; extreme weather; nitrogen; phosphorus; soil loss; subsurface runoff; surface runoff

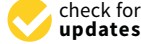



## 1. Introduction

Losses of soil and nutrients have adverse effects on the quality of soil and water. The timing, frequency and method of tillage have a major influence on erosion and soil loss [1], and soils under autumn tillage in sloping areas are more susceptible to soil and nutrient losses compared to soils with a plant cover or untilled stubble [2]. Similar results have been reported from studies conducted in Finland [3,4]. Skøien et al. [5] showed that ploughing in spring instead of autumn reduced total phosphorus (TP) losses by 50–75% on fields with high to medium erosion risk, whereas on fields with low erosion risk, changed timing had varying effects [5]. A review [2] summarized the effects of reduced and changed tillage practices on soil and phosphorus losses in a cold climate. In their study, performing tillage in autumn compared to spring increased the risk of soil and phosphorus losses [2]. Soil tillage in autumn leaves the soil bare with broken surface structures, exposing fresh, unstable surfaces to shear forces from raindrops and runoff [4]. In cold climate regions, where the growing season is short, ploughing after harvest leads to exposed soil surfaces during autumn, winter and early spring [6,7]. There is a high risk of severe soil erosion if plant cover is absent between growing seasons when most of the runoff occurs [8]. Furthermore, in some regions, most of the precipitation and hence runoff occurs during the non-growing season [8]. Several studies have shown an increase in nitrate leaching following tillage operations due to some of the aforementioned reasons [9,10].

The planting of winter cereals involves even more tillage operations in autumn compared to spring cereals. Additionally, winter cereals in a cold climate develop a sparse plant cover, which results in low protection against soil erosion. It has been shown that in a cold climate, soil and phosphorus losses are higher from winter cereals than from other areas where only autumn ploughing has been performed [11].

No tillage systems have been shown to reduce the loss of total phosphorus (TP), but some do increase the loss of dissolved reactive phosphorus (DRP) due to stratification of phosphorus in the soil [12]. Mixing the soil during spring by using spring tillage avoids stratification of soil test phosphorus, but there is a need for increased knowledge on the effect of spring tillage on DRP losses [2].

Most studies on the effects of tillage on soil and nutrient losses include only surface runoff as the main source of runoff [11]. On flat areas, however, subsurface runoff may contribute significantly to soil and nutrient losses [4,8,13]. In a cold climate, the winter conditions are important for the distribution of runoff on the different pathways. In particular, the development of snowpacks during winter can have a considerable effect on soils and runoff pathways. The temperature-regulating properties of snow are especially important for agricultural soils. Without snow cover, low temperatures can directly affect the soil and result in deep-freezing of the soil profile, which can change the soil hydraulic properties dramatically [14,15]. In a cold climate, like that of southern Norway, this has caused severe soil erosion, through impeded infiltration of snowmelt and rainwater, at the end of winter [11,16].

Climate change, which has been projected to cause higher temperatures during winter, will cause less snow to fall in low-laying areas for Norway [17], which may influence the runoff pathways, giving rise to more surface runoff and erosion. Another projection of climate change is increased intensity of precipitation. High intensity precipitation is known to cause a high risk of erosion but there is limited knowledge on the effect of tillage operations [18]. High intensity rain occurring during periods with no plant cover causes extreme erosion events [16]. Another extreme effect of climate change is the projection of more frequent summer droughts [17,19].

In a review on opportunities and challenges for Nordic agriculture in a changing climate, Wiréhn [19] identified conflicts between goals for production and eutrophication regarding the installation of drainage systems. Knowledge of the effect of soil tillage on soil and nutrient loss through surface and subsurface drainage systems during extreme weather events is a prerequisite for targeting measures.

The objective of this study was to investigate the effect of autumn tillage on surface and subsurface runoff and the loss of soil particles and nutrients from flat areas in a cold climate. Besides improving our understanding of runoff, erosion and nutrient loss processes on these areas, the experiment provided data to quantify the possible effect of extreme events on soil and nutrient losses.

## 2. Materials and Methods

### 2.1. Site Description and Experimental Design

The study site is located in Bjørkelangen, Southeastern Norway (59°53′10.5″ N 11°34′49.2″ E). The soils are dominated by silty clay loam and silty loam of marine origin. The area has an average slope of 2.5%, and according to the Norwegian erosion risk map (autumn ploughing; www.kilden.nibio.no) a potential soil loss of approximately 600 kg ha$^{-1}$ under standard management conditions. The average annual precipitation and average temperature (1981–2010) for this area are 780 mm and 5.2 °C, respectively [20].

The study site consisted of nine plots (8 m × 50 m each). All plots were tile-drained in July 2013, one year before measurements were started (1 September 2014). The results cover the five years: 2014/2015–2018/2019 (1 September to 1 September).

The soil organic matter measured using the loss on ignition for each plot varied from 4.8% to 6.3%, soil pH from 6.2 to 6.5, total phosphorus from 900 to 1300 mg kg$^{-1}$, Ammonia Lactate extractable phosphorus (P-AL) from 130 to 330 mg kg$^{-1}$ and water-extractable

($P_{H2O}$) from 3.3 to 6.6 mg kg$^{-1}$ (Table 1). The variations may reflect land management with application of more manure in the past on plot one compared to plot nine.

**Table 1.** Loss on ignition (%), pH, total phosphorus (mg kg$^{-1}$) and Ammonium Lactate extractable phosphorus (mg P-AL kg$^{-1}$) in 0–20 cm depth for the nine plots.

| Plot No. | Loss on Ignition | pH | Total Phosphorus | P-AL | Water Extractable Phosphorus |
|---|---|---|---|---|---|
| | % | | mg kg$^{-1}$ | mg kg$^{-1}$ | mg kg$^{-1}$ |
| 1 | 5.2 | 6.4 | 1300 | 336 | 6.6 |
| 2 | 4.8 | 6.4 | 1100 | 204 | 5.2 |
| 3 | 5.0 | 6.4 | 1100 | 194 | 4.1 |
| 4 | 5.1 | 6.5 | 1000 | 184 | 3.8 |
| 5 | 5.2 | 6.3 | 1000 | 140 | 3.3 |
| 6 | 5.4 | 6.3 | 1100 | 162 | 4.0 |
| 7 | 6.3 | 6.2 | 1200 | 192 | 4.6 |
| 8 | 5.7 | 6.4 | 1100 | 188 | 4.0 |
| 9 | 5.8 | 6.3 | 900 | 136 | 3.5 |

*2.2. Treatments and Agricultural Management*

The nine plots were subjected to three treatments with three replicates each year. The three treatments consisted of (a) autumn ploughing, harrowed once in spring and sown with oats (AP), (b) ploughing and harrowed three times and sown with winter wheat in autumn (WW) and (c) spring ploughing, harrowed once and sown with spring barley (SP). Each plot was in a rotation with (a) followed by (b) and (c).

The agricultural management and the soil tillage were carried out according to local farming practice of the area and with machinery available at the farm. Ploughing refers to turning the soil to a depth of 20 cm. Harrowing refers to tillage to a depth of 5–10 cms. Dates for soil tillage and sowing differs due to differences in weather (Table 2).

**Table 2.** Time of tillage and sowing for each of the three treatments in the five years (1 September–1 September).

| Treatment | Autumn Ploughing (AP) * | | Autumn Ploughing with Winter Wheat (WW) ** | | Spring Ploughing (SP) * | |
|---|---|---|---|---|---|---|
| Year | Tillage | Sowing | Tillage | Sowing | Tillage | Sowing |
| 2014/2015 | 2 September 2014 | 15 May 2015 | 2 September 2014 | 10 September 2014 | 9 May 2015 | 15 May 2015 |
| 2015/2016 | 13 October 2015 | 11 May 2016 | 12 September 2015 | 4 October 2015 | 28 April 2016 | 11 May 2016 |
| 2016/2017 | 10 October 2015 | 6 May 2017 | 6 September 2016 | 10 September 2016 | 4 May 2016 | 6 May 2017 |
| 2017/2018 | 16 October 2017 | 15 May 2018 | 22 September 2017 | 26 September 2017 | 13 May 2018 | 15 May 2018 |
| 2018/2019 | 17 October 2018 | 8 May 2019 | 3 September 2018 | 5 September 2018 | 26 April 2019 | 8 May 2018 |

\* fertilizer application together with sowing \*\* fertilizer application 16 April 2015, 21 April 2016, 7 April 2017, 3 May 2018, 26 April 2019.

Ploughing in autumn was carried out from 2 September to 17 October on the plots with spring oats (Table 2). The plots with winter wheat were ploughed, from 2 September to 22 September. The timing of spring ploughing varied from 28 April to 13 May. The sowing time for winter wheat varied between 5 September and 4 October and from 6 May to 15 May for spring cereals (Table 2). In this study, all spring cereals were sown on the same day. The amount of fertilizer applied and the yields are described in Table 3. Nutrients were applied in the form of mineral fertilizer: 110–210 kg N and 5–8 kg P ha$^{-1}$ (Table 3).

The average yields of winter wheat varied between years (1.9–7.4 tons ha$^{-1}$) with the lowest yield in 2018 when there was summer drought (Table 3). Late sowing in 2015 also caused relatively low yields of winter wheat in 2016. For barley, the average yields were 2.0–6.0 tons ha$^{-1}$ and for oats, 2.1–5.2 tons ha$^{-1}$, with the lowest yields after the dry summer in 2018. Yields of barley and oats were also low in 2018 (Table 3). Yields in winter wheat showed larger variation than yields in spring cereals (barley and oats), but were higher on average.

**Table 3.** Use of nitrogen (N) and phosphorus (P) fertilizers (kg ha$^{-1}$) and yield (tonne dry weight ha$^{-1}$).

| Treatment | Autumn Ploughing and Spring Oats (AP) | | | Autumn Ploughing and Winter Wheat (WW) | | | Spring Ploughing and Spring Barley (SP) | | |
|---|---|---|---|---|---|---|---|---|---|
| Year | Fertilizer | | Yield | Fertilizer | | Yield | Fertilizer | | Yield |
| | kg ha$^{-1}$ | | Tons ha$^{-1}$ | kg ha$^{-1}$ | | Tons ha$^{-1}$ | kg ha$^{-1}$ | | Tons ha$^{-1}$ |
| | N | P | Dry Weight | N | P | Dry Weight | N | P | Dry Weight |
| 2014/2015 | 110 | 8 | 3.7 | 130 | 5 | 7.3 | 110 | 8 | 2.6 |
| 2015/2016 | 110 | 7 | 5.2 | 110 | 4 | 4.7 | 110 | 7 | 6.0 |
| 2016/2017 | 110 | 7 | 5.1 | 160 | 6 | 7.4 | 110 | 7 | 4.9 |
| 2017/2018 | 110 | 7 | 2.1 | 210 | 7 | 1.9 | 110 | 7 | 2.0 |
| 2018/2019 | 110 | 7 | * | 180 | 7 | * | 110 | 7 | * |

* missing data due to failure of machinery.

## 2.3. Measurements

In 2014, a weather station was installed. The station measured air humidity, air temperature, precipitation in 10 min. intervals.

Annual precipitation varied from 600 mm in 2016/2017 to 836 mm 2014/2015 (Table 4). The timing of precipitation was characterized by high amounts of precipitation in autumn (September–November) except in 2016/2017 when less precipitation (approximately 50%) occurred during these months (Table 4). In September 2015, an extreme event occurred with 169 mm precipitation during this month. The year 2016/2017 showed continuously low precipitation during all months compared to the other years. During summer 2018 (May-August), precipitation was only 50% compared to the average for the other years. Average annual temperatures were between 5.8 °C and 6.2 °C. High temperatures during summer 2018 along with low precipitation resulted in a summer drought (Table 4).

**Table 4.** Measured monthly precipitation (mm) and average monthly air temperatures (°C) during the five years at Kjelle meteorological station [20].

| | Normal | Monthly Average Precipitation (mm) | | | | | Normal | Monthly Air Temperature (°C) | | | | |
|---|---|---|---|---|---|---|---|---|---|---|---|---|
| Years | 1981–2010 | 14/15 | 15/16 | 16/17 | 17/18 | 18/19 | 1981–2010 | 14/15 | 15/16 | 16/17 | 17/18 | 18/19 |
| September | 74 | 35 * | 169 | 30 | 70 | 77 | 10.1 | 10.8 * | 10.8 | 13.7 | 11 | 11.2 |
| October | 87 | 158 * | 10 | 24 | 100 | 51 | 5.4 | 8.3 * | 5.6 | 4.5 | 5.8 | 5.6 |
| November | 79 | 87 * | 62 | 61 | 94 | 88 | 0.6 | 3.4 * | 2.5 | −0.2 | −0.1 | 2.4 |
| December | 59 | 56 | 54 | 29 | 56 | 73 | −3.6 | −3.7 | 1.0 | −0.7 | −3 | −3.4 |
| January | 53 | 104 | 47 | 49 | 80 | 23 | −4.4 | −1.4 | −8.8 | −2.7 | −2.9 | −7.5 |
| February | 38 | 29 | 52 | 60 | 21 | 63 | −4.1 | −1.1 | −2.9 | −2.7 | −5.5 | −1.1 |
| March | 45 | 47 | 56 | 67 | 11 | 87 | −0.6 | 1.9 | 1.4 | 1.5 | −5.1 | 0.8 |
| April | 43 | 13 | 101 | 34 | 52 | 13 | 4.4 | 5.1 | 4.4 | 3.6 | 4.2 | 6.4 |
| May | 50 | 119 | 31 | 59 | 26 | 81 | 10.2 | 7.8 | 11.2 | 10.4 | 14.7 | 8.9 |
| June | 78 | 61 | 37 | 64 | 47 | 65 | 13.7 | 12.8 | 15.2 | 13.8 | 16.3 | 14.3 |
| July | 81 | 75 | 79 | 46 | 30 | 34 | 15.9 | 14.8 | 15.9 | 15 | 20.4 | 16.0 |
| August | 92 | 52 | 126 | 79 | 42 | 91 | 14.8 | 14.7 | 14.1 | 14.1 | 14.8 | 15.6 |
| Annual | 780 | 836 | 823 | 600 | 627 | 745 | 5.2 | 6.2 | 5.9 | 5.9 | 5.9 | 5.8 |

* from a nearby station (distance: 5 km).

A web-camera took pictures twice a day in order to observe changes on the soil surface, plant and snow cover of the plots.

Surface and subsurface discharge were measured separately with two tipping buckets for each plot (18 in total). Flow proportional water samples were taken for each tip and collected in two separate containers (36 containers in total). One container with a small hole (2.5 mm) for high discharge and one with a larger hole (8.5 mm) for low discharge. Samples were stored until there was enough water in the containers to analyze a sample.

### 2.4. Analyses of Water Samples

Water samples were analyzed for suspended sediments (SS) by filtering of a specific volume of water (between 25 and 250 mL) after thorough shaking through a preweighed filter (Whatman GF/A), the uncertainty of analysis was 15%. Total phosphorus (TP) was analyzed by the Norwegian standard method [21] with 20% uncertainty, dissolved reactive phosphorus (DRP) by the Norwegian standard method [21] with 15% uncertainty and total nitrogen (TN) by Spectrophotometry with 20% uncertainty.

### 2.5. Statistical Analysis

Data were tested and transformed to achieve normality and homogeneity of variance using RStudio. Analysis of Variance (ANOVA) was performed to test whether the differences in runoff, soil loss, losses of total phosphorus, dissolved reactive phosphorus and total nitrogen induced by treatments were statistically significant. An estimate of the least significant difference (Tukey LSD) between treatments was obtained. Statistical differences were considered significant at the $p \leq 0.05$ level.

## 3. Results and Discussion

### 3.1. Surface and Subsurface Runoff

The average total measured runoff varied from 227 mm in 2016/2017 to 728 mm in 2014/2015 (Table 5). The amount of monthly runoff differed between the five years; while the first two years showed a peak during autumn, 2017/2018 showed a similar peak in April, but no such peak occurred in 2016/2017 or 2018/2019. During the monitoring period, the highest monthly runoff (224 mm) occurred in October 2014 with high subsurface runoff (89%). High monthly runoff (187 mm) was also measured in September 2015. This event corresponded to an extreme weather event named Petra occurring in southeastern Norway [20]. Additionally, in April 2018, the total runoff was high (197 mm) and dominated by snowmelt (Figure 1).

**Table 5.** Average yearly runoff (mm) for the nine plots and the share of drainage and surface runoff.

| | | Runoff | |
|---|---|---|---|
| Year | Surface (mm) | Subsurface (mm) | Total (mm) |
| 14/15 | 171 | 557 (77%) | 728 |
| 15/16 | 112 | 414 (79%) | 525 |
| 16/17 | 80 | 146 (64%) | 227 |
| 17/18 | 85 | 384 (82%) | 468 |
| 18/19 | 26 | 304 (92%) | 330 |

The results showed that subsurface runoff was the dominating (79%) runoff pathway on this flat (2.5% slope), silty clay loam soil (Table 6). The annual subsurface runoff constituted 64–92% of the total runoff (Table 5). Correspondingly, Turtola et al. [4] found that subsurface runoff contributed to 67% of the total runoff from a flat (2% slope) clayey soil in southern Finland and in an old Norwegian study, [22] found that on average for 6 years, surface runoff amounted to 46% of the total runoff on a silty clay soil with a 4.5–9% slope. The relatively high subsurface runoff in the plot studies may be related to the installation of the tile drainage system before the establishment of each of the plot study sites [4,22]. Installation of the subsurface runoff system in 2013 in the present study site was expected to give a downward trend in subsurface runoff as the years progressed during the five-year of the study. However, no such trend was observed. In contrast, results from Turtola et al. [4] indicated a slight decrease in the share of subsurface runoff from 79% to 70% from the first (two years) to the third (five years) study period for their 10 year-time-series of data. In the first study year, subsurface runoff contributed 90% of the total runoff [4]. This indicates that subsurface runoff may be higher some years after installation of a tile drainage system.

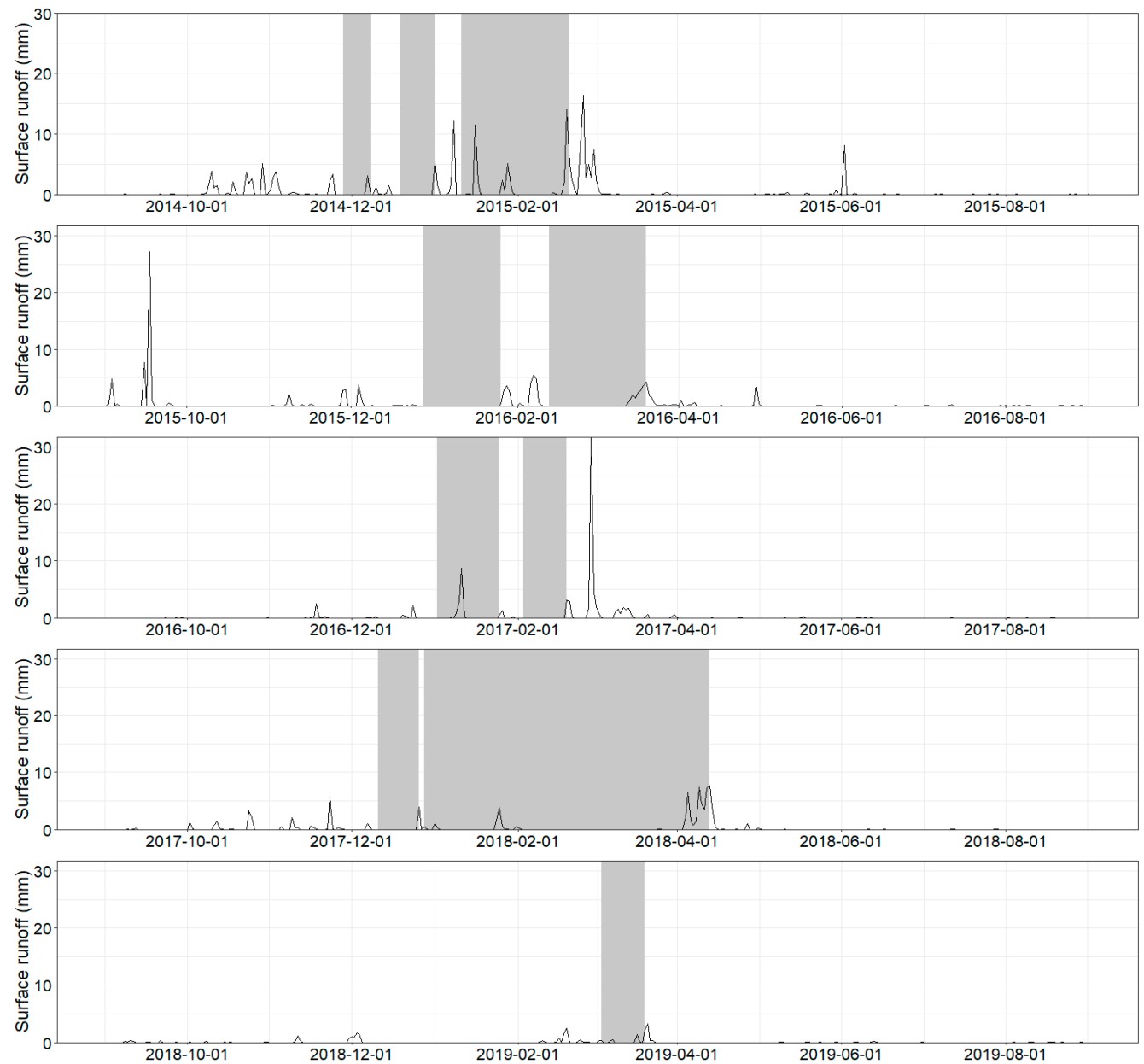

**Figure 1.** Timeseries for average surface runoff and periods with snowcover marked in gray.

Surface runoff was mainly measured during winter months (January–February) and in early spring (March and April), usually coupled with snowmelt (Figure 2). Compared to the normal period (1981–2010), temperatures were higher in February-March in the study period in all years except for 2017/2018, which contributes to increased runoff during these months (Table 4). The highest average monthly surface runoff (59 mm) occurred in February 2015, due to snowmelt and rain. At that time, the soil was partly frozen. In January 2015, high surface runoff (45 mm) occurred due to rain on frozen snow-covered soil. A reduction in infiltration capacity due to frost led to higher amounts of surface runoff in both cases. Stability of snow cover varied between years, with the least stable winter in 2014/2015 and most stable in winter 2018/2019 (Figure 1).

**Table 6.** Average annual runoff and losses of soil, total phosphorus, dissolved reactive phosphorus and total nitrogen through surface (s) and subsurface (d) runoff for autumn ploughed (AP), winter wheat (WW) and spring ploughed (SP) plots.

| Year | Tillage | Surface Runoff (mm) | Subsurface Runoff (mm) | Soil Particles (kg ha$^{-1}$) | | Total-Phosphorus (g ha$^{-1}$) | | Dissolved Reactive Phosphorus (g ha$^{-1}$) | | Total-Nitrogen (kg ha$^{-1}$) | |
|---|---|---|---|---|---|---|---|---|---|---|---|
| | | | | s | d | s | d | s | d | s | D |
| 2014/2015 | AP | 108 [a] | 651 [a] | 437 [ab] | 1726 [b] | 661 [ab] | 3670 [b] * | 126 [a] | 800 [a] | 6.7 [a] | 23.4 [b] |
| | WW | 407 [b] | 454 [a] | 1220 [b] | 808 [b] | 2174 [b] | 2205 [ab] * | 488 [b] | 533 [a] | 5.6 [a] | 9.1 [a] |
| | SP | 55 [a] | 565 [a] | 88 [a] | 557 [a] | 204 [a] | 1790 [a] * | 81 [a] | 564 [a] | 3.0 [a] | 19.5 [ab] |
| 2015/2016 | AP | 41 [a] | 474 [a] | 32 [a] | 517 [b] | 221 [a] | 1932 [b] * | 160 [a] | 629 [a] | 0.7 [a] | 14.2 [a] |
| | WW | 231 [b] | 405 [a] | 1128 [b] | 768 [b] | 2311 [b] | 2412 [ab] * | 422 [b] | 626 [a] | 6.9 [b] | 12.6 [a] |
| | SP | 99 [ab] | 361 [a] | 64 [a] | 290 [a] | 443 [a] | 1355 [a] * | 265 [ab] | 447 [a] | 1.3 [ab] | 8.2 [a] |
| 2016/2017 | AP | 59 [a] | 161 [a] | 92 [a] | 62 [b] | 192 [a] | 338 [b] * | 54 [a] | 99 [a] | 1.2 [a] | 8.4 [a] |
| | WW | 67 [a] | 146 [a] | 37 [a] | 71 [b] | 201 [a] | 386 [ab] * | 93 [a] | 147 [a] | 1.1 [a] | 7.0 [a] |
| | SP | 89 [a] | 133 [a] | 39 [a] | 84 [a] | 210 [a] | 366 [a] * | 96 [a] | 115 [a] | 1.2[a] | 8.8 [a] |
| 2017/2018 | AP | 93 [a] | 453 [a] | 116 [a] | 395 [b] | 361 [a] | 1595 [b] * | 137 [a] | 588 [a] | 1.9 [ab] | 14.8 [a] |
| | WW | 138 [a] | 277 [a] | 276 [b] | 304 [b] | 553 [b] | 1042 [ab] * | 224 [a] | 384 [a] | 2.8 [b] | 8.3 [a] |
| | SP | 31 [a] | 438 [a] | 7 [a] | 294 [a] | 157 [a] | 1404 [a] * | 129 [a] | 596 [a] | 0.7 [a] | 15.4 [a] |
| 2018/2019 | AP | 26 [a] | 312 [a] | 16 [a] | 82 [b] | 88 [a] | 867 [b] * | 41 [a] | 383 [a] | 1.6 [a] | 44.6 [b] |
| | WW | 27 [a] | 350 [a] | 12 [a] | 113 [b] | 109 [a] | 1054 [ab] * | 58 [a] | 359 [a] | 1.3 [a] | 51.7 [b] |
| | SP | 27 [a] | 254 [a] | 10 [a] | 56 [a] | 84 [a] | 603 [a] * | 49 [a] | 271 [a] | 0.8 [a] | 21.4 [a] |
| All years 2014–2019 | AP | 61 [a] | 410 [b] | 117 [a] | 557 [b] | 268 [a] | 1680 [b] * | 100 [a] | 500 [a] | 2.0 [a] | 21.1 [b] |
| | WW | 151 [b] | 323 [a] | 436 [b] | 412 [b] | 887 [b] | 1408 [ab] * | 225 [b] | 401 [a] | 3.1 [b] | 17.7 [a] |
| | SP | 55 [a] | 350 [ab] | 39 [a] | 256 [a] | 194 [a] | 1103 [a] * | 107 [a] | 398 [a] | 1.4 [a] | 14.6 [a] |

Different letters (a, b and ab) within a column represent significant differences at the 5% level of significance (LSD). * Significant level used: $p \leq 0.1$.

There was no difference in total runoff between treatments, but the distribution between surface and subsurface runoff showed differences (Table 6). On average, in the period 2014 to 2019, surface runoff was significantly ($p < 0.05$) affected by tillage practices. The winter wheat plots had significantly higher surface runoff compared to the autumn ploughed and spring ploughed plots, whereas subsurface runoff was lower for winter wheat compared to autumn ploughing (Table 6). Subsurface runoff from spring ploughed plots showed no difference compared to the other treatments. The high surface runoff from winter wheat was especially prominent during the first two study years when precipitation was highest. There was no significant difference between treatments for the years with lower precipitation. For runoff through tile drainage, which is named subsurface runoff here, the difference between treatments were too small to be significant for single years (Table 6).

The high amount of surface runoff from plots with winter wheat compared to plots with autumn and spring ploughing may be related to the extensive soil tillage, which contributes to a dense soil surface and impeded infiltration (Table 6). Skøien et al. [5] showed similar results from a plot study site in Norway where surface runoff was 10% higher for winter wheat with autumn ploughing compared to autumn ploughing and spring ploughing. They argued that increased tillage operations through ploughing and harrowing before sowing in winter wheat plots led to more soil compaction, reduced volume of drainable pores and reduced infiltration rates. Indication of reduced infiltration was also observed in December 2014–February 2015 in the present study when increased surface runoff and reduced subsurface runoff were measured from winter wheat plots compared to the other treatments, indicating lower infiltration from winter wheat. According to the conceptual model from Rittenburg et al. [23], surface runoff occurs for soils that have low organic matter show surface crust formation, are fine-textured, are degraded with decreased macroporosity or are frozen. High precipitation during autumn 2014 after several tillage operations may have contributed to surface crust formation, decreased macroporosity and

surface runoff for winter wheat plots. Additionally, the winter 2014/2015 was unstable with and without snow cover and several large runoff events occurred (Figure 1). The amount of surface runoff from autumn ploughed and spring ploughed plots showed no significant difference in the present study (Table 6). Skøien et al. [5] showed that plots with spring ploughing had the lowest surface runoff in nine out of 11 years in one site, whereas in another site, spring ploughing resulted in more surface runoff compared to autumn ploughing. These differences were due to differences in soil structure and soil texture.

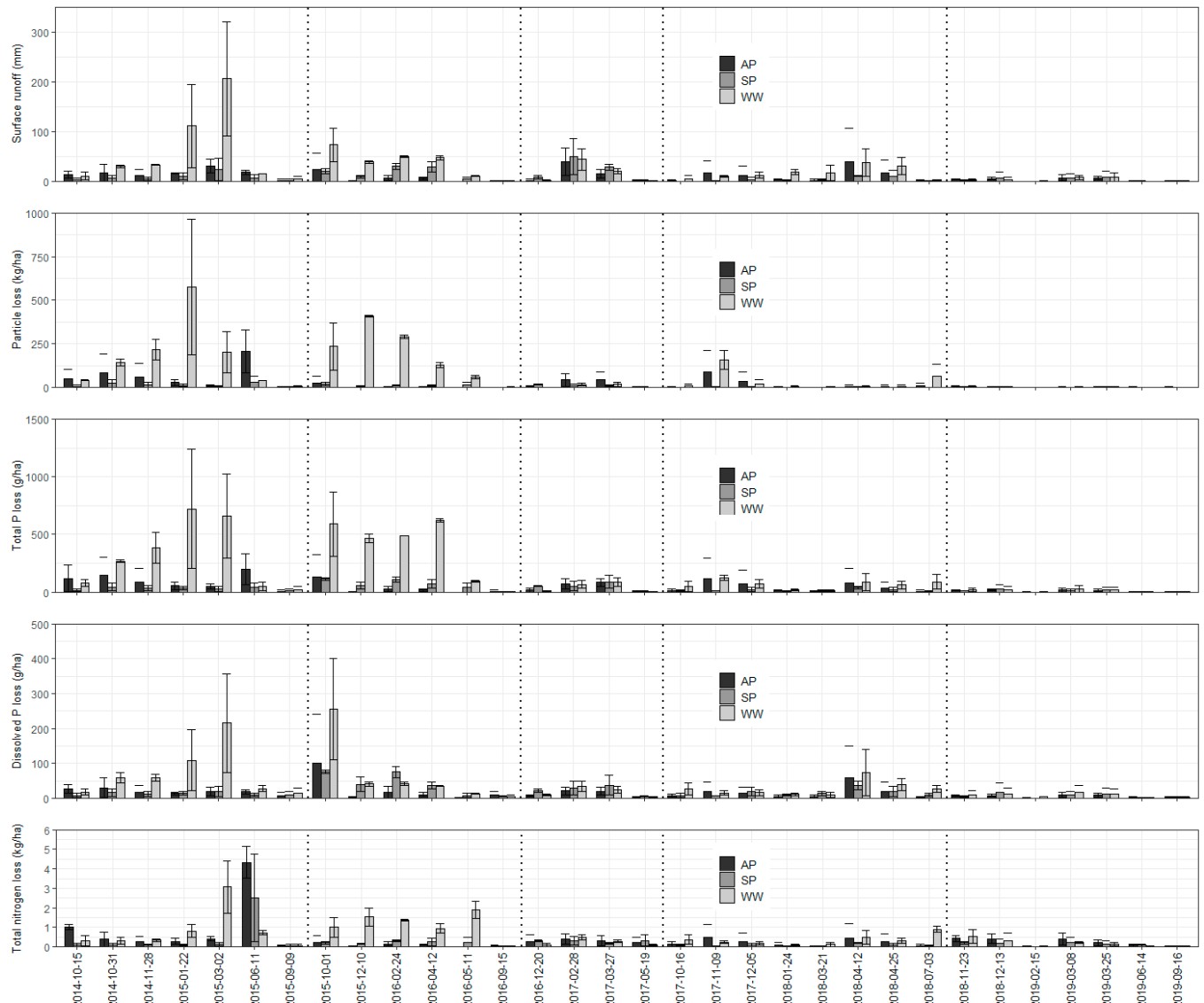

**Figure 2.** Averages values for each tillage practice (AP, WW, SP) for surface loss of soil and nutrients, measured surface runoff (mm), precipitation (mm) and the dates when the samples were taken. Years are separated by dashed lines.

Subsurface runoff was higher from plots that were only ploughed compared to those with ploughing and winter wheat (Table 6). Subsurface runoff is a result of infiltration and is limited by the hydrological conductivity of the subsurface soil or the existence of preferential flow pathways from the surface to the tile drains [23]. Our results indicate that the flow pathways from surface to tile drains were less restricted for ploughed compared to winter wheat plots. The high subsurface runoff from ploughed plots was observed for most of the timeseries, except 2018/2019. In 2018/2019, after the very dry summer in 2018, the first runoff occurred in November. Surface runoff was also low during winter 2019 for

all treatments. High subsurface runoff from ploughed plots was especially pronounced in February 2015 and January 2018 (Figure 3). This is explained by the high macroporosity and water storage capacity in the plough layer, with more time to infiltrate compared to the winter wheat with restricted surface infiltration [4].

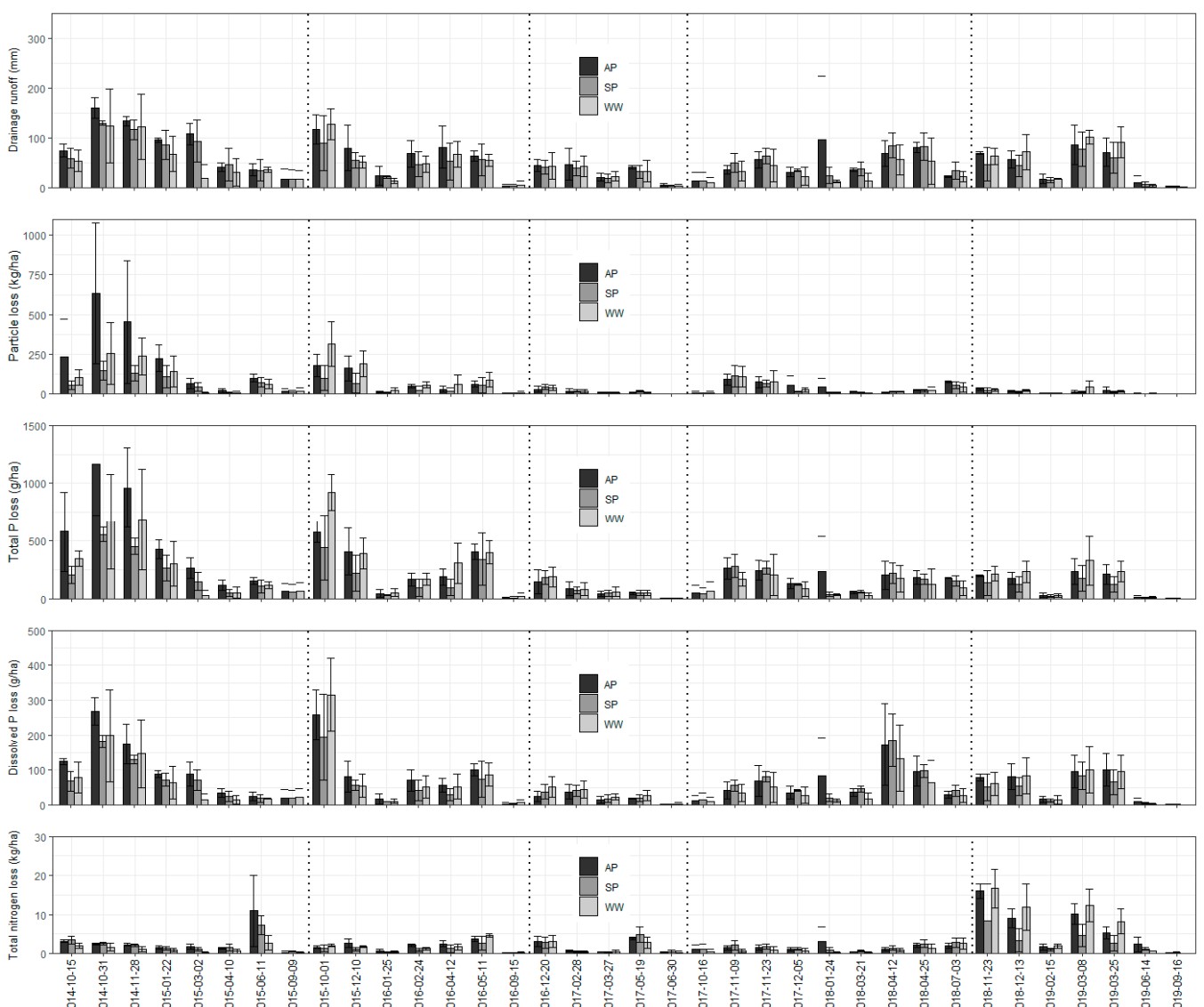

**Figure 3.** Averages values for each tillage practice (AP, WW, SP) for soil and nutrient losses through tile drainage, measured subsurface runoff (mm), precipitation (mm) and the dates when the samples were taken. Years are separated by dashed lines.

### 3.2. Loss of Soil Particles

Soil losses were highest in 2014/2015 and 2015/2016 when precipitation and runoff were highest (Table 6).

Overall, there was a significant ($p < 0.01$) effect of tillage practice on soil loss. Loss of soil particles through subsurface runoff was lower from the spring ploughed treatment (256 kg SS ha$^{-1}$) compared to autumn ploughing (557 kg SS ha$^{-1}$) and winter wheat (412 kg SS ha$^{-1}$) on average for all years and also for all single years (Table 6). In surface runoff, on average for all years, the loss of soil particles was higher (436 kg SS ha$^{-1}$) from winter wheat compared to both autumn ploughed (117 kg ha$^{-1}$) and spring ploughed plots (39 kg ha$^{-1}$) (Table 6). The high soil losses from winter wheat may have been reduced

by direct drilling of winter wheat instead of ploughing before planting [11]. However, the great advantage of spring tillage regarding soil loss may be a disadvantage for soil structure in some years. Tilling in spring causes increased risk of compaction when soil moisture is above field capacity, as is common in spring [24].

The high loss of soil particles in surface runoff from winter wheat was especially prominent for the two first years with high precipitation and runoff (Figure 2). Only once in 2014/2015 did soil losses in surface runoff surpass losses in subsurface runoff on the winter wheat. The high surface losses mainly occurred in the period between 28 November and 22 January (Figures 2 and 3). The importance of alternating snow cover and snowmelt for loss of soil particles during winter has been shown by [25]. Ulén [25] argues that concentrations of soil particles increase at the end of a snowmelt episode. Thus, the concentration of soil particles in runoff is expected to be higher in years with frequent periods with snowmelt than in years with fewer snowmelt events. The number of snowmelt events during winter 2014/2015 was higher than for other years (Figure 1). In this period, runoff was triggered mainly by short periods of accumulation of snow followed by melt together with rain (Figure 1). During these times the top layer of the soil was frozen, which partially prohibited infiltration. Soil loss in surface runoff during the unstable winter was 140 kg SS ha$^{-1}$ (28 November 2014–2 March 2015) on average for all treatments. In contrast, winter 2017/2018 was relatively stable, indicated by the observed continuous snow cover (Figure 1). During the stable winter, soil loss in surface runoff was 2 kg SS ha$^{-1}$ (5 December 2017–21 March 2018) on average for all treatments. Starkloff [16] showed in a similar climate that major soil erosion was caused by a small rain event on frozen ground before snow cover was established, while snowmelt played no significant role in terms of soil erosion in their study.

In 2014 and 2015, autumn losses of soil particles were high. Total runoff during September and October 2014 was 223 mm and the corresponding soil loss was 591 kg SS ha$^{-1}$ on average for all treatments. In September 2015, after ploughing of the winter wheat plots, an extreme meteorological event (Petra) with high intensity and amounts of precipitation occurred in southeastern Norway. At the plot study site, approximately 76 mm of precipitation was measured during the five days (14–18 September 2015) of the Petra extreme event. From 9 September to 1 October 2015, the soil loss was 547 kg SS ha$^{-1}$, corresponding to 29% of the total annual soil loss, which was measured from winter wheat plots (Figures 2 and 3). Extreme weather with intense precipitation has been shown to cause severe soil erosion. Bechmann [8] reported 5 to 6 times higher soil losses in July than the average of 11 years due to heavy rainfall in July. Lundekvam and Skøien [26] reported greater soil loss on autumn ploughed than spring ploughed plots after a heavy rainstorm (80 mm in 1 h) in June 1995.

In 2015/2016, surface runoff was the main source of soil loss on the winter wheat plots. Moreover, the surface runoff occurred mainly in autumn and winter that year (Figure 2). Frozen soil reduced infiltration from January to April 2016, which also led to higher losses in surface runoff compared to subsurface runoff.

Loss of soil particles occurred on average mainly (67%) through subsurface runoff (Table 6). Mitigating soil erosion through subsurface runoff is a challenge in this region, whereas in other regions, surface runoff is the main pathway for soil particles (examples can be found in [27,28]. The present study showed that on relatively flat silty clay, loam soils subsurface runoff can be an important pathway for soil particles (Table 6) and are in agreement with studies in Finland [4,13]. As mentioned for runoff, this could be influenced by installation of the tile drainage system. For winter wheat, surface and subsurface soil loss contributed equally to the total soil loss, whereas for spring, ploughing surface soil loss constituted only 13% of total soil loss.

### 3.3. Loss of Total Phosphorus

Loss of total phosphorus occurred, as for soil particles, mainly (average 76%) through subsurface runoff (Table 6). The highest losses occurred during the two first years and especially during autumn 2014; a similar situation occurred for soil particles (Figure 3).

In surface runoff, there was a close link between losses of particulate phosphorus and losses of soil particles (PP = 0.0015 SS; $R^2$ = 0.98) and correspondingly for subsurface runoff (PP = 0.0018 SS; $R^2$ = 0.82) (Figure 4). According to these relationships, the content of particulate phosphorus in soil particles in surface runoff was 1.5 g P kg$^{-1}$ soil and in subsurface runoff 1.8 g P kg$^{-1}$, which is higher than the measured content of total phosphorus in soil (0.9–1.3 mg P kg$^{-1}$; Table 1). This indicates an enrichment of phosphorus in both surface and subsurface runoff [29].

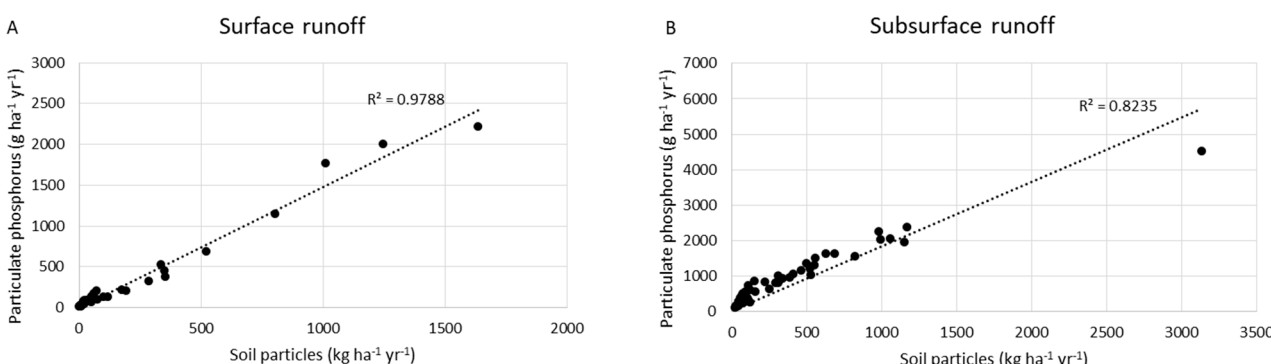

**Figure 4.** Relationship between annual soil (kg ha$^{-1}$) and particulate phosphorus loss (g ha$^{-1}$) for surface and subsurface runoff from single plots.

Overall, there was a significant effect of tillage practice on phosphorus losses (Table 6). Higher losses of total phosphorus were measured in surface runoff from winter wheat (887 g TP ha$^{-1}$) compared to autumn ploughing (268 g TP ha$^{-1}$) and spring ploughing (194 g TP ha$^{-1}$). For winter wheat, the total phosphorus losses in surface runoff during spring 2015 and 2016 were relatively high compared to the loss of soil particles (Figure 2).

Total phosphorus losses through subsurface runoff were lower from spring ploughed plots compared to autumn ploughed plots, but the difference between treatments were less pronounced for total phosphorus compared to soil particles (lower significance-level). For winter wheat, the losses of total phosphorus through subsurface runoff did not differ from the other treatments (Table 6).

Even though there is a good relationship between losses of particles and total phosphorus, there were some deviating trends. In 2016/2017, the spring ploughed plots suddenly showed higher losses of total phosphorus compared to autumn ploughed plots in both surface (not significant) and subsurface runoff (Table 6). However, highest particle losses occurred on the autumn ploughed plots in surface runoff. The reasons for this could be that this year had a relatively dry autumn period compared to the first two years (Figures 2 and 3), which allowed for an early establishment of the winter wheat. Furthermore, both years were characterized by 200 mm less precipitation compared to the other years. The highest loss of soil particles and total phosphorus through subsurface runoff was from autumn ploughed plots one year after the tile drainage system was installed (Figure 4).

Loss of dissolved reactive phosphorus occurred on average mainly (75%) through subsurface runoff (Table 6). Loss of dissolved reactive phosphorus in surface runoff showed similar results as for soil particles and total phosphorus with higher losses in surface runoff from winter wheat (225 g DRP ha$^{-1}$) compared to autumn ploughing (100 g DRP ha$^{-1}$) and spring ploughing (107 g DRP ha$^{-1}$). There was no difference in dissolved reactive phosphorus losses through subsurface runoff between treatments (Table 6). A similar experiment in Finland showed higher DRP losses from no-till compared to ploughing [12].

This was, according to the authors, possibly due to the development of a conductive pore structure from soil surface to drain depth [12]. In the present study, the soil was ploughed each year either in autumn or spring and no such direct transport pathways developed.

The soil P status differed between the nine plots. The highest soil P status was measured in plot 1, while the lowest was measured in plot 9 (Table 1). The soil P status showed an effect ($R^2$ = 0.5) on DRP/TP in surface runoff (Figure 5). The main process of total phosphorus losses was related to soil loss, but the soil P status was an additional factor causing increased loss of dissolved reactive phosphorus. Svanbäck et al. [30] reported no significant difference in dissolved reactive phosphorus leaching between conventional ploughing and shallow tillage, although shallow-tilled plots had a slightly higher P-AL (n.s.). However, the level of P-AL was much lower (32 mg P-AL $kg^{-1}$) in the study by [30] compared to the soil P status of the present study (136–336 mg P-AL $kg^{-1}$; Table 1).

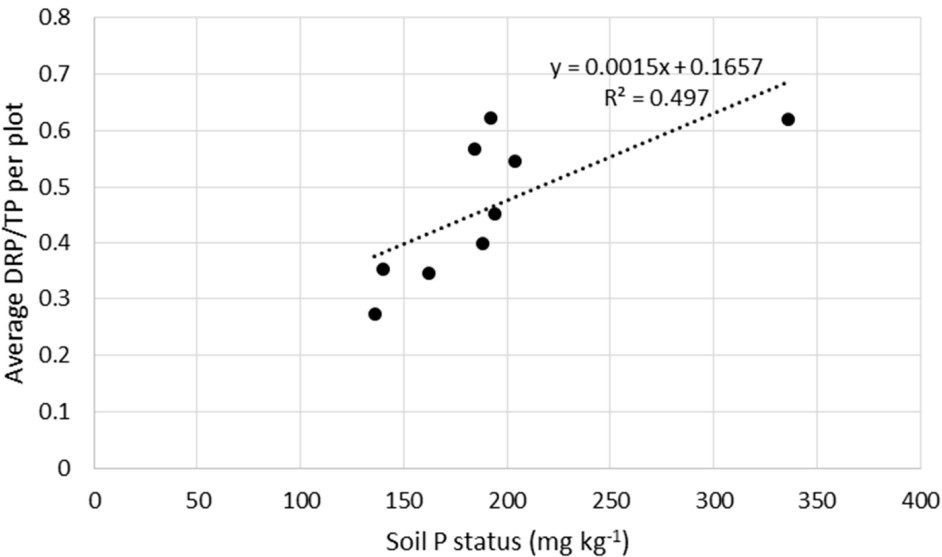

**Figure 5.** Average dissolved reactive phosphorus (DRP)/total phosphorus (TP) in surface runoff all years per plot in relation to soil P status (mg P-AL $kg^{-1}$).

Due to the close relationship between loss of soil particles and total phosphorus, the effects of weather conditions during winter as described for soil particles are also applicable to total phosphorus. Severe erosion and loss of total phosphorus occur during events with rain on frozen or snowcovered soil [11]. Hoffman et al. [31] showed that late winter snowmelt and mixed snowmelt and rain events increased total phosphorus losses and the DRP/TP ratios with a need to focus management actions on this period.

### 3.4. Loss of Total Nitrogen

The loss of total nitrogen occurred on average mainly (89%) through subsurface runoff (Table 6). The importance of subsurface runoff as a transport pathway for nitrogen has been shown by [32]. In a Norwegian study, nitrogen concentrations in subsurface runoff were 2–4 times higher than the concentrations in surface runoff [33]. The amount of subsurface runoff was also much higher than surface runoff in the latter study.

The growing season 2018 was dry and the yields of barley and oats were approximately halved compared to earlier. The yields of winter wheat were lower than 30% of average yields for earlier years (Table 3). Due to the low yields combined with fertilizer application at a normal level, there was a high surplus of nitrogen in the soil. The losses of total nitrogen after the dry summer in 2018 (the study year 2018/2019) were 40 kg TN $ha^{-1}$, more than twice the average total nitrogen losses for other years (15 kg TN $ha^{-1}$) (Table 6). Salo and Turtola [34] showed in an example from 1999 that the surplus of nitrogen due to drought was higher from cereals than for grass. Rankinen et al. [35] found in their

simulations that 1 kg surplus of nitrogen in production corresponded to 0.3 kg nitrogen leaching. This is in line with the present study; however, major variations due to tillage were also found.

For autumn ploughed plots, losses of total nitrogen through subsurface runoff were significantly ($p < 0.05$) higher compared to both spring ploughed and winter wheat plots (Table 6). Furthermore, ploughing in autumn 2018 resulted in more than a doubling of total nitrogen losses compared to not ploughing in autumn (spring ploughing) (Table 6). This occurred although nitrogen surplus was much higher on the plots that were left unploughed in autumn 2018. Due to rotation, the plots with winter wheat in 2017/2018 were followed by spring ploughing in 2018/2019 and the nitrogen surplus was much higher for plots with winter wheat compared to oats and barley (Table 3). The results clearly showed that despite the higher surpluses of nitrogen in plots with winter wheat during summer 2018 compared to other treatments, the losses of total nitrogen from these plots (not ploughed in autumn 2018) were lower (21.4 kg TN ha$^{-1}$) compared to the plots with oats (51.7 kg TN ha$^{-1}$) and barley (44.6 kg TN ha$^{-1}$) during summer 2018 and followed by tillage in autumn (oats were followed by winter wheat and barley followed by autumn ploughing). Thus, results show that ploughing, rather than nitrogen surplus, was more important for total nitrogen losses.

In 2015, the majority of total nitrogen losses occurred during spring (Figures 2 and 3). One single runoff event after tillage and sowing caused high losses of total nitrogen.

Losses of total nitrogen in surface runoff were significantly ($p < 0.05$) lower from spring and autumn ploughed plots compared to winter wheat plots (Table 6). Loss of total nitrogen in surface runoff was very high for winter wheat during the sampling period 10 April–11 June 2015, possibly due to runoff of surface-applied fertilizer (Table 2). Fertilizer application in the spring and autumn ploughed plot were also carried out during this period, but were applied below the soil surface (Table 2). In total, there were higher losses in surface runoff from winter wheat (3.1 kg TN ha$^{-1}$) compared to autumn ploughing (2.0 kg TN ha$^{-1}$) and spring ploughing (1.4 kg TN ha$^{-1}$).

## 4. Conclusions

Subsurface runoff was the main source for soil (67%), total phosphorus (76%), dissolved reactive phosphorus (75%) and total nitrogen (89%) losses on this flat, loamy soil of marine origin under cold climate conditions. Most runoff (79%) occurred through the subsurface runoff. Through the subsurface pathway, lowest soil losses resulted from spring ploughed plots. Losses of total phosphorus through subsurface runoff were also lower from spring ploughing compared to autumn ploughing. There was no difference in losses of dissolved reactive phosphorus in subsurface runoff between the treatments. However, the total nitrogen losses were higher from autumn ploughing compared to other treatments. Most losses of soil and phosphorus occurred in autumn and winter when runoff was highest. Total nitrogen losses occurred mainly during autumn and spring. Summer drought caused high losses of nitrogen the following seasons, mainly on plots with autumn tillage. Highest soil and nutrient losses coincided with the highest runoff events, indicating that climate change with increased precipitation and increased runoff during autumn and winter increases the risk of high losses in the future. However, the present study shows that spring tillage compared to autumn tillage has the potential to reduce soil and nutrient losses through subsurface pathways, especially under extreme weather conditions.

**Author Contributions:** Data collection, data analysis, writing—original draft preparation and writing were done in cooperation between the two authors. Review and editing were done by main author. All authors have read and agreed to the published version of the manuscript.

**Funding:** The present study was funded by the Norwegian Agriculture Agency.

**Institutional Review Board Statement:** Not applicable.

**Informed Consent Statement:** Not applicable.

**Data Availability Statement:** Data is contained within the article.

**Acknowledgments:** This investigation is based on data obtained from the Kjelle plot study in cooperation with Kjelle vgs. The study design was developed and carefully maintained by Geir Tveiti.

**Conflicts of Interest:** The authors declare no conflict of interest.

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
