# Peer review of "Soil Tillage and Crop Growth Effects on Surface and Subsurface Runoff, Loss of Soil, Phosphorus and Nitrogen in a Cold Climate"

_land, doi:10.3390/land10010077_

Round 1

Reviewer 1 Report

Soil tillage and crop growth not only affect surface runoff, but also affect subsurface runoff. This paper studies the effects of soil tillage and crop growth on the loss of surface and subsurface runoff, as well as soil nitrogen and phosphorus, which is innovative.

However, there are still several problems after reading the article:

  1. The introduction to the literature review is slightly insufficient, please strengthen the review content of the research literature;

  1. Strengthen the description of the study area, and supplement related conditions such as farming conditions, soil properties, surface runoff, and subsurface runoff;

  1. Extreme weather has an important impact on surface runoff and subsurface runoff. How does the article deal with the impact of different weather conditions on surface runoff and subsurface runoff? Please elaborate in the text.

  1. The research results lack comparative analysis, please add relevant content.

Author Response

Thank you for the review and comments to the manuscript. I have responded to your comments here below.

Best regards,

Marianne Bechmann

Point 1: The literature review in the introduction has been strengthened; more literature has been included and the discussion improved.

Point 2: The description of the study area includes soil type, slope and climate. The farming activities has been described more in detail with for example number of tillage operations and depth of tillage. Surface and subsurface runoff has been described in the result section.

Herewith Attached is the revised manuscript

Point 3. The different weather conditions has been discussed in the text, exemplified by extreme events that occurred during the study period.

Point 4. The discussion has been improved by including a more comparative analysis of the results with reference to newer relevant litterature (Uusitalo et al. 2018, Hoffmann et al. 2019, Wiréhn 2018).

Reviewer 2 Report

Dear author's,

Presented manuscript present very important theme, especially in the light of negative trend in application of huge amount of nitrogen (especially in autumn). Personally, I am not fan of ploughing, but I am very interested about new research in that field and cognitions which arise from it.

I will recommended manuscript for publication, but you need to make some corrections before.

My comments and suggestions you may find in attached pdf.

Author Response

Thank you very much for the review with comments in the manuscript. We have revised the manuscript according to your suggestions. It helped improve the manuscript. I herewith attach the revised manuscript.

Best regards

Marianne Bechmann

Reviewer 3 Report

This is a well written manuscript that presents important findings about surface and subsurface runoff with associated soil and nutrient losses from cold region crops.  Suggested changes are relatively minor but must be addressed prior to acceptance.

The description of treatments needs to be improved.  By referring to 2 treatments as autumn plowing, this implies that the tillage is the same but after reading it is clear that the tillage on the two treatments was not the same.  Detailed description of the intensity and implements used, as well as timing, must be included for all three treatments.  Timing of fertilization for the 3 treatments should be clarified.  Author might be discussed why 2 different spring crops were selected, confounding the tillage timing and crop effects for those 2 treatments. 

Given the greater environmental impacts of the WW treatment, the authors should discuss options to reduce the erosion and nutrient losses from this crop.  Because it is higher yielding, producers will want to continue planting WW.  However, can the authors suggest mitigation strategies such as reduced tillage (which has been successful in many regions, including cold regions) or changes in the fertilization formulations or timing or crop establishment timing?

The loss of soil by subsurface flow is quite different results from most regions.  While other studies in this region have found similar results, the authors could highlight this finding more clearly.  It poses challenges to mitigating erosion when the flow is subsurface.

Additional comments in the annotated file attached.

Author Response

Thank you very much for the review with comments in the manuscript. I have revised the manuscript according to your suggestions. It helped improve the manuscript. A few questions, did not lead to changes in the text, but I will give the answers here below.

Regarding farming practice. The tillage depth has now been described more in detail with specified depth of tillage and timing of all tillage methods used. We chose to use oats, barley and winter wheat in the different treatments to avoid any pest due to having the same crop year after year. This was done due to an advice by the farmer in the study area. Looking back, we could have chosen only one spring and one winter crop. However, the crop and the growing season has minimum effect on the runoff and losses of soil and nutrients measured and therefore have minmum influence on the results. Also fertilizer application was done according to the level used by the farmer the single years. The point was to make the experiment as close to farming practice as possible. A possible method, namely direct drilling, to reduce soil erosion from winter wheat has been suggested in the text. The challenge regarding soil losses through subsurface drainage has been highlighted.

Comment to line 340: Regarding the soil test P values (P-AL) we did analyse these at the start of the experiment. The five years of this experiment is too short time to see any changes in soil test P and therefore timeseries of these data are not presented.

I herewith attach the revised manuscript.

Best regards,

Marianne Bechmann

Reviewer 4 Report

I recommend the publication of this manuscript. The methods are well-described and thorough and the results are well presented too. A minor comment, please do keep in mind the reason why spring tillage is not done in many places (increased risk of compaction when soil moisture is above field capacity, as is common in spring). This point should be made in the discussion when you advocate for spring tillage compared to autumn tillage

Author Response

Thank you very much for the review. We agree on your comment and have included a sentence on the problem with spring tillage in the discussion.

Best regards

Marianne Bechmann